# Design and Experimentation of a Self-Sensing Actuator for Active Vibration Isolation System with Adjustable Anti-Resonance Frequency Controller

**DOI:** 10.3390/s21061941

**Published:** 2021-03-10

**Authors:** Yuan Fu, Shusen Li, Jiuqing Liu, Bo Zhao

**Affiliations:** 1School of Mechanical and Electrical Engineering, Northeast Forestry University, Harbin 150080, China; fuyuan@nefu.edu.cn (Y.F.); lishusenzp@126.com (S.L.); 2Key Lab of Ultra-Precision Intelligent Instrumentation (Harbin Institute of Technology), Ministry of Industry and Information Technology, Harbin 150080, China; hitzhaobo@hit.edu.cn

**Keywords:** self-sensing actuator, active vibration isolation, self-sensing controller, anti-resonance frequency

## Abstract

The vibration isolation system is now indispensable to high-precision instruments and equipment, which can provide a low vibration environment to ensure performance. However, the disturbance with variable frequency poses a challenge to the vibration isolation system, resulting in precision reduction of dynamic modeling. This paper presents a velocity self-sensing method and experimental verification of a vibration isolation system. A self-sensing actuator is designed to isolate the vibration with varying frequencies according to the dynamic vibration absorber structure. The mechanical structure of the actuator is illustrated, and the dynamic model is derived. Then a self-sensing method is proposed to adjust the anti-resonance frequency of the system without velocity sensors, which can also reduce the complexity of the system and prevent the disturbance transmitting along the cables. The self-sensing controller is constructed to track the variable frequency of the disturbance. A prototype of the isolation system equipped with velocity sensors is developed for the experiment. The experiment results show that the closed-loop transmissibility is less than −5 dB in the whole frequency rand and is less than −40 dB around, adding anti-resonance frequency which can be adjusted from 0 Hz to initial anti-resonance frequency. The disturbance amplitude of the payload can be suppressed to 10%.

## 1. Introduction

With the continuous improvement of measurement and manufacturing accuracy, micro vibration isolation technology has become one of the common core technologies in the field of high-precision manufacturing, processing, and measurement [1,2,3,4]. In the semiconductor manufacturing field represented by a lithography machine [5], ultra-precision machining field represented by a diamond lathe [6], ultra-precision measurement field represented by a super-resolution fluorescence microscope and gravitational wave detection [7,8], a precision vibration isolation system is needed to provide stable and reliable environmental protection. Therefore, in-depth study of vibration isolation technology is of great practical value to ensure the accuracy of the system. 

Stiffness control is one of the key unit technologies of a vibration isolation system [9,10,11,12]. If the positive and negative stiffness mechanisms are combined reasonably, the system characteristics such as zero stiffness and infinite stiffness can be obtained. Mizuno of Saitama University in Japan has studied that the zero-power magnetic levitation system can effectively suppress the direct load disturbance by connecting the zero-power magnetic levitation system with the traditional spring in series to obtain infinite stiffness [13]. Los realized the negative stiffness structure by parallel connection of the compression bar and spring, which greatly reduced the natural frequency of the system in the vertical direction [14]. Zhou et al. designed an adjustable electromagnetic isolator with high static and low dynamic stiffness. The mechanical spring and electromagnet are connected in parallel to form a magnetic spring structure, and the stiffness of the magnetic spring can be adjusted by controlling the current magnitude and direction of the electromagnet [15]. Melik-Shakhnazarov studied a dynamic modeling and decoupled control of miscellaneous active vibration isolation systems designed for laboratory (workshop), transport, and space applications [16]. The parameters of the dynamic blocks of the system and the design characteristics can be determined when the active frequency range and maximal vibration attenuation factor were given, and the factors which limit the efficacy bounds of vibration isolators were analyzed. 

In the lithography system, as shown in Figure 1, the function of the vibration isolator is to isolate the environmental vibration, which can ensure the accuracy of the measurement system. The reticle state and wafer stage can move on the base frame. The measurement frame on which the measurement system and objective lens are mounted is supported by the vibration isolators. There are two kinds of environmental vibration acting on the base frame: one is the vibration from the environment, the other is the vibration caused by inertia force which is generated by the acceleration and deceleration movement of the reticle state and wafer stage. The vibration from the environment is with low frequency, and the vibration from inertia force is with specific frequency related to trajectory planning. Moreover, during the system dynamics modeling, the reticle state and wafer stage are controlled by sweep frequency signal, respectively, to obtain the dynamic parameters. The inertial force generated during frequency sweeping acts on the platform base, which is equivalent to the disturbing force varying with frequency acting on the whole system, and directly affects the measurement accuracy of the optical system, then leading to poor model accuracy. The existing vibration isolators cannot effectively isolate the disturbance with frequency variation, especially for the vibration signal between 0 Hz and natural frequency. Therefore, it is necessary to propose a new vibration isolation control method to meet the requirements of frequency conversion disturbance signal suppression. Furthermore, the cables of the velocity sensors mounted on the measurement frame are connected to the control system which is placed on the ground, so the environmental vibration may transmit from the ground to the measurement frame through the cables. 

Self-sensing technology has attracted more and more interest in reducing the complexity of the system. Freyer designed a hardware-in-the-loop system which consisted of a self-sensing actuator with adaptive feedback control for vibration suppression of an emulated tool, and the dual functionality of the piezoelectric transducer was verified [17]. Chen designed a monolithic self-sensing stage for active vibration isolation [18]. The self-sensing actuator can both generate high-resolution displacement and monitor the dynamic characteristics of the stage. Yan proposed a self-sensing electromagnetic transducer for vibration control of a space antenna reflector [19]. The system did not require a separate sensor or a real-time controller, so the equipment was lightweight and the controller was simplified. Dong investigated the vibration characteristics of a sandwich structure, which is made of graphene-reinforced composite and piezoelectric ceramics [20]. Ju proposed an improved vibration suppression system using the piezoelectric self-sensing technique [21]. A single piezoelectric element was employed as an actuator and a sensor simultaneously. Pelletier [22] and Qiu [23] applied a piezoelectric self-sensing actuation in active vibration control in the same way. The piezoelectric transducer (PZT) can be also used as a vibration damper [24,25]. Al-Bahrani studied the self-sensing properties of nanocomposites based on multi-walled carbon nanotubes (MWCNTs) [26]. A micro-scale model was proposed to predict the variation of the resistance caused by the micro-indentation damage. Brugo integrated the composite laminate with a nanostructured piezoelectric sensor based on PVDF-TrFE nanofibers and aluminum sheets as electrodes [27]. The real-time electrical response of the self-sensing laminate was investigated by non-destructive impact tests, and the performance were evaluated in terms of linearity and spatial uniformity.

In this paper, a self-sensing method for active vibration isolation system with adjustable anti-resonance frequency is proposed. The self-sensing controller built can both replace the velocity sensors on the measurement frame and isolate the disturbance with frequency variation. The mechanical structure of the self-sensing actuator is designed, and the dynamic model of the vibration isolation system is established. The effectiveness of the self-sensing method is validated by both simulation and experiment. This paper is organized as follows. In Section 2, the self-sensing active vibration isolation system is designed and the dynamic model is built. At the same time, the design of the controller and the parameters calculation process is given. Experiment verification is provided in Section 3. Finally, the conclusions are drawn in Section 4.

## 2. Materials and Methods

### 2.1. Description and Mathematical Modeling of the System

#### 2.1.1. Design of the Self-Sensing Active Vibration Isolation System

Based on the analysis of the dynamic vibration absorption principle, the prototype of self-sensing active vibration isolation system is designed as shown in Figure 2, including the payload, the base, the connecting plate, the air floating rails, the spring, the voice coil motor (VCM), as well as the self-sensing PZT actuator. The VCM has long motion stroke with zero stiffness. However, it only works in low-frequency band and its resolution is not high enough for micro vibration isolation. The self-sensing PZT has the advantages of wide band and extremely high resolution, but the stroke of the PZT is only a few tens of microns. The self-sensing active vibration isolation system combines the advantages and obtains wide control frequency band, high resolution, long motion stroke, and self-sensor.

The schematic of the self-sensing active vibration isolation system is shown in Figure 3. The payload is modeled as a rigid body with the mass *m*_1_, which is directly driven by the force of PZT *f_p_*. The mover of the VCM, the PZT, and the support frame are connected together with the total mass *m*_2_, which are driven by the force of VCM *f_v_*. The support frame is supported by the spring and guided by the air bearing, with the stiffness *k*_1_ and the damping *c*_1_. The payload position *x*_1_ is disturbed by the environmental vibration *x_b_*, and *x*_2_ is the displacement of the mover of VCM. In the active control system, the velocity sensor located on the base is used to measure the absolute velocity of the base *v*_1_. The absolute velocity of the payload *v*_2_ is indirectly measured by the self-sensing method in this paper. 

#### 2.1.2. Design of Self-Sensing PZT Actuator

The self-sensing PZT actuator is a crucial component of a self-sensing active vibration isolation system as it not only provides active ways to achieve position control and vibration control, but also acts as sensors of the proposed system. The schematic of the self-sensing PZT actuator is shown in Figure 4. The self-sensing PZT actuator is a crucial component of the active vibration isolation system. It not only provides active force for vibration control, but also acts as a velocity sensor. The schematic of the self-sensing PZT actuator is shown in Figure 4. The force output rod can connect to the payload with the screw thread, and is guided by the linear bearing to move along the direction of the PZT force. The self-sensing PZT is stressed by the pre-tightening spring to compensate for the gap between the front baffle and the rear baffle. The self-sensing PZT is a multilayer stack which consists of thin piezoelectric sheets. It can be seen that the total height of the self-sensing is 21.4 mm. The material types of sensor and actuator are PZT-8 and PZT-5H, respectively. The prototype of the self-sensing PZT actuator is shown in Figure 5. 

The self-sensing PZT actuator is provided a certain amount of preload by the pre-tightening spring. In this case, all the piezoelectric materials are used in *d*_33_-operation where the “three”-direction refers to the direction of polarization. It implies that for the self-sensing PZT actuator, we are interested in its abilities of actuating and sensing along the poling axis. The piezoelectric effect is an effect in which energy is converted between mechanical and electrical form. When a pressure is applied to the self-sensing PZT actuator, the resultant mechanical deformation will result in an electrical charge. The generated charge is amplified by a charge amplifier that has a certain capacitance *C*. Thus, the generated voltage *Vs* of the piezoelectric force sensor due to the load force *f_p_* can be calculated as,
(1)$$V_{s}=\frac{d_{33}f_{p}}{C}$$

In order to reduce any unexpected vibration, active vibration control methods have to integrate sensors, controllers, power amplifiers, and actuators into the system. These additional external devices increase both the weight and the cost, and reduce the reliability and safety of the system. At the same time, the collection of the payload displacement needs an extra speed sensor. In order to avoid the base vibration to the payload caused by the cable, we designed a transfer function to use *Vs* as a feedback signal of the controller reflecting and replacing the payload displacement information. 

#### 2.1.3. Mathematical Modeling of the Self-Sensing Active Vibration Isolation System

The output force of the VCM follows the Faraday’s law,
(2)$$f_{v}=NB\pi dI=k_{i}I$$
where *N* is the number of coils turns, *B* is the average magnetic induction of the air gap, *d* is the diameter of coil, *I* is the current value of the coil, and *k_i_* = *NB*π*d*, is the thrust coefficient of the VCM.

One-dimension piezoelectric constitutive equation is as follows
(3)$$\{\begin{array}{c} D=d_{33}\sigma +\varepsilon_{33}E\\ \sigma =E_{p}(\varepsilon -d_{33}E) \end{array}$$
where *D* is the longitudinal electric displacement, *d*_33_ is the longitudinal piezoelectric strain coefficient, *σ* is the longitudinal stress of the piezoelectric ceramic piece, $\varepsilon_{33}$ is the dielectric constant, *E* is the longitudinal electric field intensity, *E_p_* is the elastic modulus, ε is the longitudinal strain.

The operating voltage, longitudinal displacement, and output force of the self-sensing PZT can be derived as
(4) U=Eh
(5)δ=nεh
(6)$$f_{p}=-A\sigma =k_{u}V_{a}-k_{p}\delta$$
where *h* is the thickness of single layers of stacked actuator, δ is the total deformation of stacked ceramic layers, *n* is the number of stacked ceramic layers, *A* is the area of piezoelectric ceramic piece, *k_u_* = *AE_p_d*_33_/*h* is the force coefficient of the PZT, *V_a_* is the voltage of the self-sensing PZT actuator, *k_p_* = *AE_p_*/(*nh*) is the stiffness of the PZT.

Because the stiffness of piezoelectric ceramics is too large, it is reduced by connecting PZT in series with the spring. The stiffness of the PZT actuator can be rewritten as
(7)$$k_{p}=(AE_{p}k_{L}/(nh))/(AE_{p}/(nh)+k_{L})$$

The equations of motion of the self-sensing active vibration isolation system are given by
(8)$$\{\begin{array}{l} m_{2}{\ddot{x}}_{2}=f_{v}-f_{p}\\ m_{1}{\ddot{x}}_{1}=f_{p}-k(x_{1}-x_{b})-c({\dot{x}}_{1}-{\dot{x}}_{b}) \end{array}$$

By substituting (2) and (6) into (8), and noting $\delta =x_{1}-x_{2}$, the dynamic equations become
(9)$$\{\begin{array}{l} m_{2}{\ddot{x}}_{2}=k_{i}I-k_{u}V_{a}+k_{p}(x_{1}-x_{2})\\ m_{1}{\ddot{x}}_{1}=k_{u}V_{a}-k_{p}(x_{1}-x_{2})-k(x_{1}-x_{b})-c({\dot{x}}_{1}-{\dot{x}}_{b}) \end{array}$$

#### 2.1.4. Characteristic Analysis

It is a linear time-invariant (LTI) system and it is assumed that the initial condition of the system is zero. Let control inputs become zero, namely, *I* = 0 and *V_a_* = 0. Then the Laplace transform of (9) can be written as
(10)$$\{\begin{array}{l} m_{2}s^{2}X_{2}(s)=k_{p}X_{1}(s)-k_{p}X_{2}(s)\\ m_{1}s^{2}X_{1}(s)=(-cs-k_{p}-k)X_{1}(s)+k_{p}X_{2}(s)+(cs+k)X_{b}(s) \end{array}$$

The transfer function from *x_b_* to *x*_1_ is used to evaluate the vibration isolation performance. The inherent transfer function can be derived from (10) as
(11)$$\frac{X_{1}(s)}{X_{b}(s)}=\frac{\left(m_{2}cs^{3}+m_{2}ks^{2}+k_{p}cs+kk_{p}\right)}{\left[m_{1}m_{2}s^{4}+m_{2}cs^{3}+(m_{1}k_{p}+m_{2}k+m_{2}k_{p})s^{2}+k_{p}cs+kk_{p}\right]}$$

The value of *c* is small enough that it can be neglected. In order to express the frequency transfer characteristics of the self-sensing active vibration isolation system more clearly, Equation (11) is equivalent to
(12)$$\begin{array}{ll} \frac{X_{1}(s)}{X_{b}(s)} & =\frac{m_{2}ks^{2}+kk_{p}}{m_{1}m_{2}s^{4}+(m_{1}k_{p}+m_{2}k+m_{2}k_{p})s^{2}+kk_{p}}\\ & =\frac{k}{m_{1}}\frac{s^{2}+\omega_{3}^{2}}{\left(s^{2}+\omega_{1}^{2})(s^{2}+\omega_{2}^{2}\right)} \end{array}$$
where $\omega_{1}\omega_{2}=\omega_{3}\sqrt{k/m_{1}}$, $\omega_{1}^{2}+\omega_{2}^{2}=k_{p}/m_{2}+k/m_{1}+k_{p}/m_{1}$ and $\omega_{3}=\sqrt{k_{p}/m_{2}}$.

Figure 6 shows inherent transfer characteristics of system. ω_1_ and ω_2_ are two resonant frequencies of the system. When the frequency of the base vibration is around them, the vibration is amplified. ω_3_ is the anti-resonance frequency of the system. When the frequency of the base vibration is around it, the vibration there is isolated entirely. This characteristic of the system is known as anti-resonance. However, once the system is determined, the initial anti-resonance frequency is fixed. If the anti-resonance frequency can be adjusted in the real time, the self-sensing active vibration isolation system can achieve better vibration isolation performance.

### 2.2. Design of the Self-Sensing Controller

#### 2.2.1. Design of the Adjustable Anti-Resonance Frequency Controller

According to Figure 6, the design principle of controller consists of two parts. One is to suppress the extra vibration, whose frequency is around two resonant frequencies, ω_1_ and ω_2_, and the other is to add a new anti-resonance frequency, ω_4_, which is adjustable by tracking the external disturbance in the real time. Base on the above ideas, a closed-loop control block diagrammatic sketch is designed as shown in Figure 7.

The difference of the expected value, *x_r_*, (*x_r_* = 0), and the actual value of the payload displacement, *x*_1_, is used as the input of the controller, and the output is the current value of the coil, *I*, and the voltage value of the self-sensing PZT actuator, *V_a_*. The main frequency of the base vibration, *f*, is obtained by FFT (Fast Fourier Transform), which is to determine the newly added anti-resonance ω_4_. With the input of *I* and *V_a_*, the driver produces the voice coil motor force, *f_v_*, and the self-sensing PZT stack force, *f_p_*, to act on the payload. Above all, there forms a closed-loop control to isolate the base vibration.

Even if there exists more efficient controller design methodology, since the focus of the paper is using self-sensing instead of a speed sensor for our dual-stage actuator system in the context of practical application, we chose to use a rather simple and intuitive graphical tuning methodology for controller design. The design of the controller contains two transfer functions from *x*_1_ to *I* and from *x*_1_ to *V_a_*, namely *H*_1_ and *H*_2_, designed as follows
(13)$$H_{1}(s)=\frac{I(s)}{X_{1}(s)}=K+\frac{1}{\tau s}$$
(14)$$H_{2}(s)=\frac{V_{a}(s)}{X_{1}(s)}=\frac{10^{4}}{s^{2}+\omega_{4}^{2}}$$
where *K*, τ, ω_4_, are adjustable parameters of the designed controller.

By comparing the transmissibility curves of simulation and experiment, the damping is small enough that it can be ignored. Then the Laplace transform of (9) can be written as
(15)$$\{\begin{array}{l} m_{2}s^{2}X_{2}(s)=k_{i}I(s)-k_{u}V_{a}(s)+k_{p}X_{1}(s)-k_{p}X_{2}(s)\\ m_{1}s^{2}X_{1}(s)=k_{u}V_{a}(s)-(k_{p}+k)X_{1}(s)+k_{p}X_{2}(s)+kX_{b}(s) \end{array}$$

Substituting (13) and (14) to (15) yields the expression of the transfer function of the system with the designed controller.
(16)$$\frac{X_{1}(s)}{X_{b}(s)}=\frac{ks(s^{2}+\omega_{3}^{2})(s^{2}+\omega_{4}^{2})}{a_{1}s^{7}+a_{2}s^{5}+a_{3}s^{3}+a_{4}s^{2}+a_{5}s+a_{6}},$$
where ω_3_ and ω_4_ are the initial anti-resonance frequency and the added anti-resonance frequency as defined in Section 2.1.3 and Section 2.2,
$$\begin{array}{l} a_{1}=m_{1}m_{2},a_{2}=m_{1}k_{p}+m_{2}k_{p}+m_{2}k+m_{1}m_{2}\omega_{4}^{2},\\ a_{3}=m_{1}k_{p}\omega_{4}^{2}-10^{4}\cdot m_{2}k_{u}+m_{2}k_{p}\omega_{4}^{2}-k_{p}k_{i}K+m_{2}k\omega_{4}^{2}+kk_{p},a_{4}=-k_{p}k_{i}/\tau ,\\ a_{5}=kk_{p}\omega_{4}^{2}-k_{p}k_{i}K\omega_{4}^{2},a_{6}=-k_{p}k_{i}\omega_{4}^{2}/\tau . \end{array}$$


Figure 8 shows the conceptual bode diagram of system with the designed controller. Compared with Figure 6, the proposed controller suppresses the extra vibration, whose frequency is around ω_1_ or ω_2_. The purpose of the proposed control method is to adjust anti-resonance frequency according to the external disturbance, and give full play to the unique advantages of the dynamic vibration absorption.

#### 2.2.2. Design of Self-Sensing Transfer Function

Based on the ideas mentioned in Section 2.1.2, a new closed-loop control block diagrammatic sketch is designed as shown in Figure 9. *x*_1*s*_ is the payload position obtained by self-sensing when *H*_3_ is introduced. It is equivalent to *x*_1_.

The dynamic formulas about *f_p_* are follows,
(17)$$\{\begin{array}{l} m_{2}{\ddot{x}}_{2}=f_{v}-f_{p}\\ f_{p}=k_{u}V_{a}-k_{p}(x_{1}{}_{s}-x_{2})\\ f_{p}=CV_{s}/d_{33} \end{array}$$

Then the Laplace transform of (8) can be written as,
(18)$$\{\begin{array}{l} m_{2}s^{2}X_{2}(s)=k_{i}I(s)-k_{u}V_{a}(s)+k_{p}(X_{1}{}_{s}(s)-X_{2}(s))\\ k_{u}V_{a}(s)+k_{p}(X_{1}{}_{s}(s)-X_{2}(s))=CV_{s}(s)/d_{33} \end{array}$$

By substituting (2) and (6) into (18), transfer function, *H*_3_, can be derived as,
(19)$$H_{3}(s)=\frac{X_{1s}(s)}{V_{s}(s)}=\frac{C[m_{2}\tau s^{5}+(m_{2}\omega_{4}^{2}+k_{p})\tau s^{3}+k_{p}\omega_{4}^{2}\tau s]}{d_{33}[m_{2}k_{p}\tau s^{5}+(m_{2}k_{p}\tau \omega_{4}^{2}-10^{4}m_{2}k_{u}\tau -k_{i}k_{p}K\tau )s^{3}-k_{i}k_{p}s^{2}-k_{i}k_{p}K\tau \omega_{4}^{2}s-k_{i}k_{p}\omega_{4}^{2}]}$$

Figure 10 shows the simulation comparison between the payload displacement *x*_1_ and *x*_1*s*_ obtained by the speed sensor and the self-sensing PZT, respectively, when the base vibration is a step disturbance. The consistency of the simulation results shows that the self-sensing scheme is feasible and the theoretical derivation is correct.

#### 2.2.3. Parameters Setting of the Self-Sensing Controller

The designed controller contains three parameters, *K*, τ, ω_4_. The value of parameter ω_4_ equals added anti-resonance frequency. It is up to the main frequency of the base vibration to decide, $\omega_{4}=2\pi f$. 

Selecting state variables $x={\left[x_{2}\;{\dot{x}}_{2}\;x_{1}\;{\dot{x}}_{1}\right]}^{T}$, control variables $u={\left[I\;V_{a}\right]}^{T}$, and disturbances $w={\left[x_{b}\;{\dot{x}}_{b}\right]}^{T}$, the absolute speed of the load can be measured by the speed sensor as the measurement output, that is $y={\dot{x}}_{1}$, and the state space description of the system can be obtained
(20)$$\left\{ \begin{array}{l} \dot{x}=Ax+B_{1}w+B_{2}u\\ y=Cx \end{array}\right.$$
where $A=\left(\begin{array}{cccc} 0 & 1 & 0 & 0\\ -\frac{k_{p}}{m_{2}} & 0 & \frac{k_{p}}{m_{2}} & 0\\ 0 & 0 & 0 & 1\\ \frac{k_{p}}{m_{1}} & 0 & -\frac{k+k_{p}}{m_{1}} & -\frac{c}{m_{1}} \end{array}\right)$, $B_{1}=\left(\begin{array}{cc} 0 & 0\\ 0 & 0\\ 0 & 0\\ \frac{k}{m_{1}} & \frac{c}{m_{1}} \end{array}\right)$, $B_{2}=\left(\begin{array}{cc} 0 & 0\\ \frac{k_{i}}{m_{2}} & -\frac{k_{u}}{m_{2}}\\ 0 & 0\\ 0 & \frac{k_{u}}{m_{1}} \end{array}\right)$, $C={\left[0\;0\;0\;1\right]}^{T}$.

Through the analysis of Lyapunov stability criterion, the constraint conditions of system stability are obtained. The eigenvalues of matrix A all have sub real parts. Once the controller is tuned, stability is verified for posterity by checking the eigenvalues of the global system state space matrix. The optimal solution selection principle is to minimize the closed-loop gain on the premise of satisfying the stability constraints. The optimum solutions of *K* and τ are as follows.
(21)$$\begin{array}{l} \begin{array}{cc} \underset{K,\tau }{\min} & G_{\max} \end{array}\\ s.t.\{\begin{array}{c} |I|<I_{\max}\\ |V_{a}|<V_{a}{}_{\max} \end{array}, \end{array}$$
where *G*_max_ is the maximum value of closed-loop gain, *I*_max_ is the maximum of the current value of the VCM, *V_a_*_max_ is the maximum of the voltage value of the self-sensing PZT actuator.

The *G*_max_ can be derived as
(22)$$G_{\max}=\sup[20\mathrm{lg}(\left|H(j\omega )\right|)]$$
(23)$$|H(j\omega )|=\frac{k\omega (\omega_{3}^{2}-\omega^{2})(\omega_{4}^{2}-\omega^{2})}{\sqrt{{\left(a_{6}-a_{4}\omega^{2}\right)}^{2}+{\left(-a_{1}\omega^{7}+a_{2}\omega^{5}-a_{3}\omega^{3}+a_{5}\omega \right)}^{2}}}$$

The transfer function from *v_b_* to *V_a_* can be written as
(24)$$\frac{V_{a}(s)}{V_{b}(s)}=\frac{10^{4}ks(s^{2}+\omega_{3}^{2})}{a_{1}s^{7}+a_{2}s^{5}+a_{3}s^{3}+a_{4}s^{2}+a_{5}s+a_{6}}$$

The maximum of the gain of *v_b_* to *V_a_*, |*G_Va_*|_max_, can be derived as
(25)$$\left|G_{V_{a}}{|}_{\max}=\sup(\left|H_{V_{a}}(j\omega )\right|\right)$$
(26)$$\left|H_{V_{a}}(j\omega )\right|=\frac{10^{4}\left|k\omega (\omega_{3}^{2}-\omega^{2})\right|}{\sqrt{{\left(a_{6}-a_{4}\omega^{2}\right)}^{2}+{\left(-a_{1}\omega^{7}+a_{2}\omega^{5}-a_{3}\omega^{3}+a_{5}\omega \right)}^{2}}}$$

Three different values τ are selected and set as 0.1 × 10^−4^, 0.2 × 10^−4^, 0.3 × 10^−4^. Then, the three relationship curves of *G*_max_ and |*G_Va_*|_max_ with *K* are obtained when each τ is fixed.

Figure 11 shows the relationship curves of *G*_max_ and |*G_Va_*|_max_ with *K*. We can get that, with the decrease of *K*, *G*_max_ is also reduced. *G_Va_*_max_ is the saturation value of |*G_Va_*|_max_. We can also get the relationship curves of |*G_Va_*|_max_ and *K* are same under different τ. The intersection point of each curve with the saturation, *K*_min_, is the desirable minimum value of *K* under the corresponding τ. From Figure 11, we can get that, with the decrease of *K*, |*G_Va_*|_max_ increases on the contrary.

Therefore, *K* cannot be infinitely reduced due to the limitation of the range of the control signal *V_a_*.

The transfer function from the velocity of the base vibration, *v_b_* to *I*, can be written as
(27)$$\frac{I(s)}{V_{b}(s)}=\frac{k(K\tau s+1)(s^{2}+\omega_{3}^{2})(s^{2}+\omega_{4}^{2})}{b_{1}s^{7}+b_{2}s^{5}+b_{3}s^{3}+b_{4}s^{2}+b_{5}s+b_{6}}$$
where $b_{1}=\tau a_{1}$, $b_{2}=\tau a_{2}$, $b_{3}=\tau a_{3}$, $b_{4}=\tau a_{4}$, $b_{5}=\tau a_{5}$, $b_{6}=\tau a_{6}$.

The maximum of the gain of *v_b_* to *I*, |*G_I_*|_max_, can be derived as
(28)$$\left|G_{I}{|}_{\max}=\sup(\left|H_{I}(j\omega )\right|\right)$$
(29)$$\left|H_{I}(j\omega )\right|=\frac{\left|k\omega (\omega_{3}^{2}-\omega^{2})(\omega_{4}^{2}-\omega^{2})\right|\cdot \sqrt{{\left(K\tau \omega \right)}^{2}+1}}{\sqrt{{\left(b_{6}-b_{4}\omega^{2}\right)}^{2}+{\left(-b_{1}\omega^{7}+b_{2}\omega^{5}-b_{3}\omega^{3}+b_{5}\omega \right)}^{2}}}$$

Three different *K* values are selected and set as 200, 400, 600. Then, the three relationship curves of *G*_max_ and |*G_I_*|_max_ with τ are obtained when each *K* is fixed.

Figure 12 also shows the relationship curve of |*G_I_*|_max_ and |*G_I_*|_max_ with τ. We can get that with the decrease of τ, *G*_max_ is also reduced. *G_I_*_max_ is the saturation value of |*G_I_*|_max_. The intersection point of each curve with the saturation, $\tau_{\min}$, is the desirable minimum value of τ under the corresponding *K*. From Figure 12, we can get that, with the decrease of τ, |*G_I_*|_max_ increases on the contrary. 

Therefore, τ cannot be infinitely reduced due to the limitation of the range of the control signal *I*. 

In this paper, the permissible control current range of the VCM is −1.67 to 1.67 A, the permissible control voltage range of the PZT is 0 to 150 V, and the maximum speed of the base vibration is 7 mm/s. Through the above analysis, the selection process of the parameters of the proposed controller is as follows. At first, determine the anti-resonance frequency, the value of ω_4_ to be added according to the main frequency of the base vibration. Then, based on Equation (21) and the conclusion from Figure 11 and Figure 12, the optimum solution of *K* and τ can be chosen properly for the designed controller for best performance of the self-sensing active vibration isolation system.

## 3. Results and Discussion

Figure 13 shows the experiment setup of the self-sensing active vibration isolation system, whose parameters are listed in Table 1. The velocities of the base and the payload are measured by two velocity sensors (GS-11D type). The natural frequency of the velocity sensor is reduced from 4.5 Hz to 0.18 Hz by using frequency expansion technology to measure the low frequency vibration. The VCM is driven by the TA115 type power amplifier. The supply voltage of TA115 is 24 V to 48 V. The equivalent motor voltage is up to ±43 V. The control signal voltage range of TA115 is 0–10 V and the gain is 0.2 A/V to 0.8 A/V. In the experiment, we set the gain to be 0.2 A/V. The PZT is driven by RH31 type actuator. The control voltage range of RH31 is 0–5 V and the gain is 30.69 through the experiment calibration. The load velocity, base velocity, and the self-sensing voltage are collected by the real-time target system. The close-loop controller of the experiment system is realized by the real-time target system as well, and the system state data can be visualized on the computer. This section will provide a concise and precise description of the experimental results, their interpretation, as well as the experimental conclusions that can be drawn.

### 3.1. Experiment Verification

#### 3.1.1. Experiment Verification of Frequency Domain

The external vibration is generated from hitting the base by a pulse hammer, GK-2110, and is detected and analyzed by a 941B type vibration analyzer. The inherent transmissibility of the self-sensing active vibration isolation system is measured as shown in Figure 14. Two resonance and one anti-resonance frequencies are approximately 7.9 Hz, 17.4 Hz, 13.8 Hz. Figure 15 indicates that the resonance peaks are eliminated with the proposed controller. Furthermore, the transmissibility is less than −5 dB in the whole frequency band, less than −20 dB when the frequency is less than the initial anti-resonance, and less than −40 dB when the frequency is around the added anti-resonance. With the change of the controller parameter, the added anti-resonance frequency of the system changes at the same time. The experiment results are consistent with the theoretical design of the self-sensing active vibration isolation system with the proposed controller.

#### 3.1.2. Experiment Verification of Self-Sensing Active Vibration Isolation System

The external vibration is generated from hitting the base by a Japan pulse hammer, GK-2110. The velocities of the base and the payload are measured by the GS-11D type speed sensor. Based on the FFT result of the external vibration, the proposed controller sets ω_4_ = 15, *K* = 150, and τ = 0.00001.

Figure 16 infers that the proposed controller achieves a good performance on isolating the external disturbance. Without any controller, the base vibration velocity is 7 mm/s, the maximum velocity of the payload vibration is 5 mm/s, the payload continues to oscillate, and the vibration attenuation is slow. With the proposed controller, the payload vibration velocity decays more rapidly and the maximum velocity of payload vibration is 0.5 mm/s with a reduction of 90.0% compared to the uncontrolled.

Figure 17 shows that the vibration isolation performance of the whole system during the platform base is applied with the sweep frequency disturbance, which is generated by a vibration exciter. Set 10 Hz, 20 Hz, 30 Hz, 40 Hz, and 50 Hz as sweeping frequencies, the amplitude of each frequency disturbance is set to the same, 5 mm/s, and each frequency disturbance continues for 2.0 s. When the sweep frequency of the disturbance changes, the FFT predictor will adjust the parameters in real time to realize the anti-resonance frequency point tracking. The results show that the reduction at 10 Hz, 20 Hz, 30 Hz, 40 Hz, and 50 Hz are 91.4%, 93.1%, 91.2%, 92.0%, and 94.0%, respectively, compared with the uncontrolled.

Figure 18 shows the comparison of vibration performance between the feedback signal obtained by the speed sensor and the self-sensing, respectively, when the base vibration is generated by a pulse hammer. The experiment results are consistent with the simulation results, which verifies that the self-sensing method can also achieve the same or even better vibration isolation performance as using the velocity sensor. Moreover, without the velocity sensors, the whole vibration isolation system and the environmental vibration are simplified.

Figure 19 shows that, by following the principle of selecting the parameters of the controller proposed in Section 2.2.3, the control signals *I* and *V_a_* are not saturated, which is in accordance with the expected requirement of the experiment, and verified the rationality of the design of the self-sensing controller.

## 4. Conclusions

In this paper, a self-sensing controller for a vibration isolation system with anti-resonance frequency is proposed. The mechanical structure of the actuator is illustrated, and the dynamical model of the isolation system is built. The self-sensing controller is proposed to isolate the vibration without velocity sensors on the payload based on the dynamical model. The controller is validated by both simulation and experiment. By comparing the simulation results and the experiment results, the self-sensing method for the vibration isolation system is verified, and the controller is proved to be effective.

First, the use of the self-sending actuator prevents the disturbance transmitting along the cables, as well as reduces the complexity of the system. Then, the system can adjust the anti-resonance frequency point in the real time by tracking the sweep frequency disturbance and achieve good vibration isolation performance. Furthermore, the disturbance amplitude of the payload decays from 5 mm/s to 0.5 mm/s with a reduction of 90.0% for the impulse disturbance applied to the platform base. So, the velocity sensors can be replaced by the self-sensing controller, and the self-sensing method can be applied in the active vibration isolation system, especially at frequency sweeping in the semiconductor manufacturing industry.

## Figures and Tables

**Figure 1 sensors-21-01941-f001:**
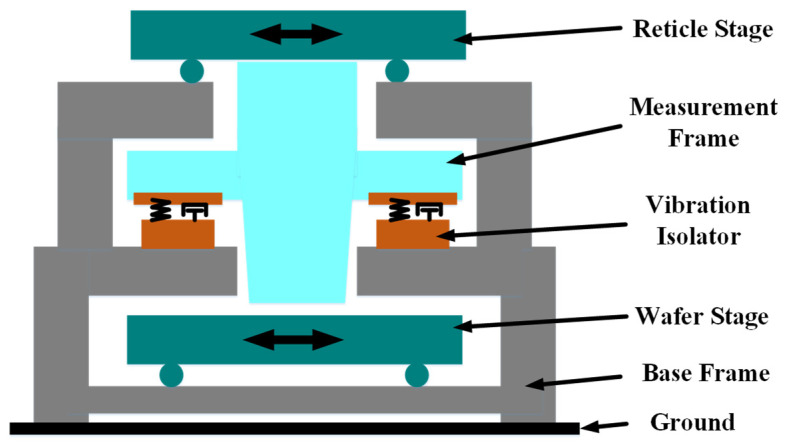
Basic structure of lithography system.

**Figure 2 sensors-21-01941-f002:**
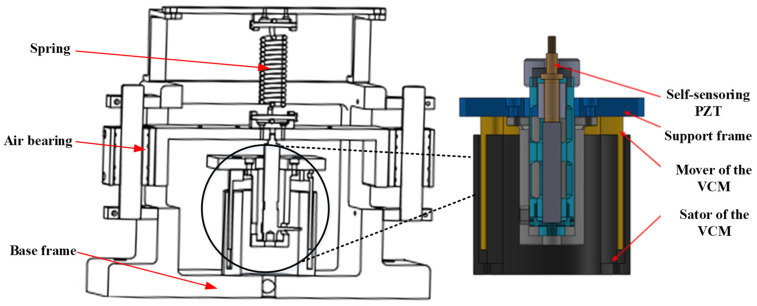
The prototype of self-sensing active vibration isolation system.

**Figure 3 sensors-21-01941-f003:**
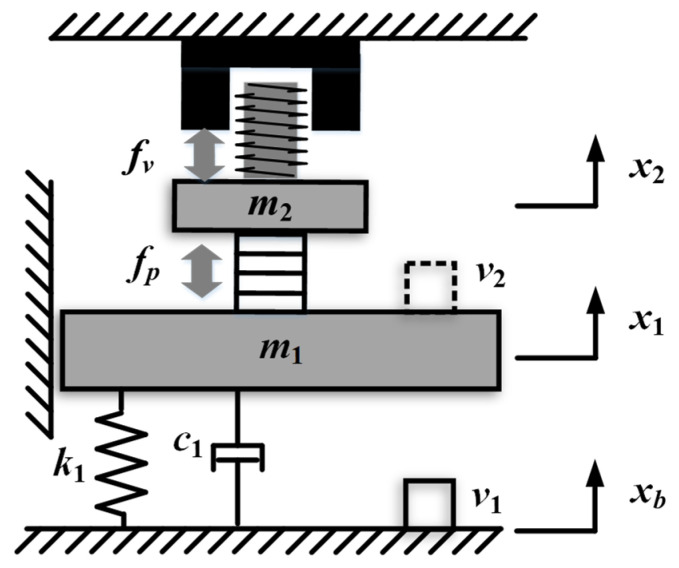
The schematic of the self-sensing active vibration isolation system.

**Figure 4 sensors-21-01941-f004:**
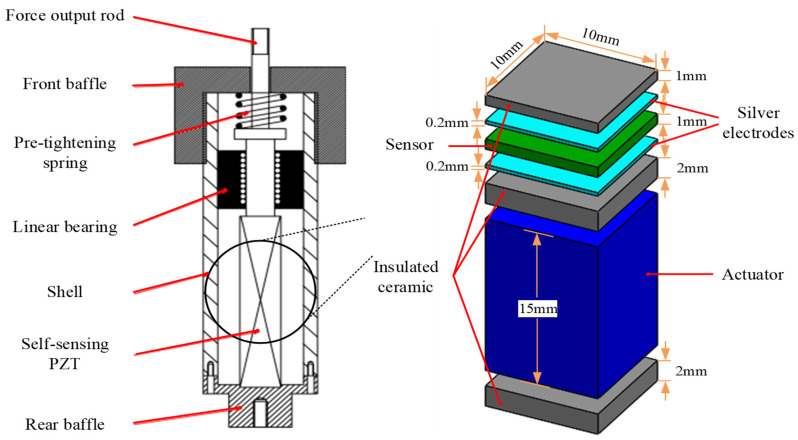
Schematic of the self-sensing piezoelectric transducer (PZT) actuator.

**Figure 5 sensors-21-01941-f005:**
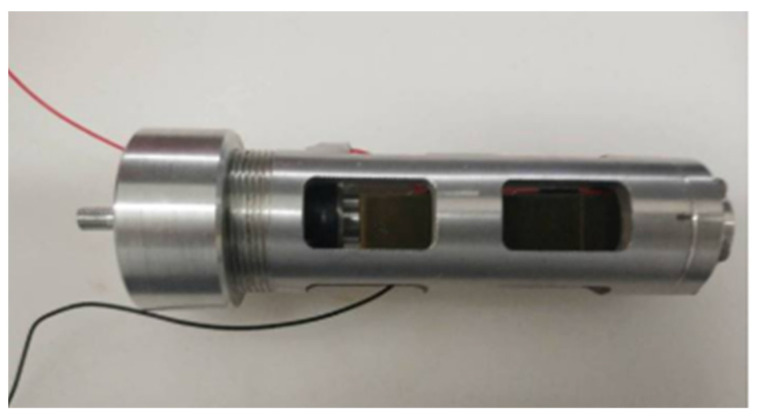
The prototype of the self-sensing PZT actuator.

**Figure 6 sensors-21-01941-f006:**
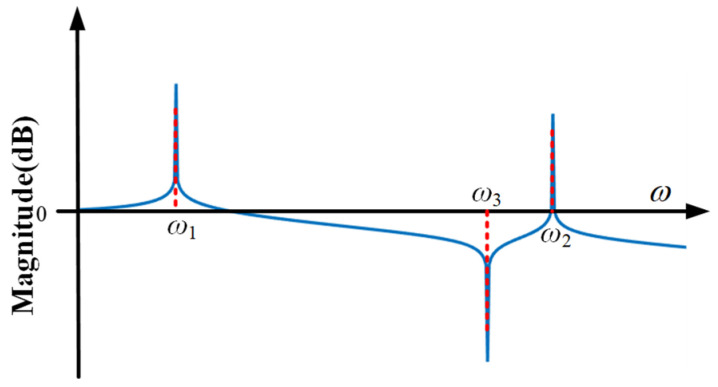
Bode diagram of the system without any controller.

**Figure 7 sensors-21-01941-f007:**
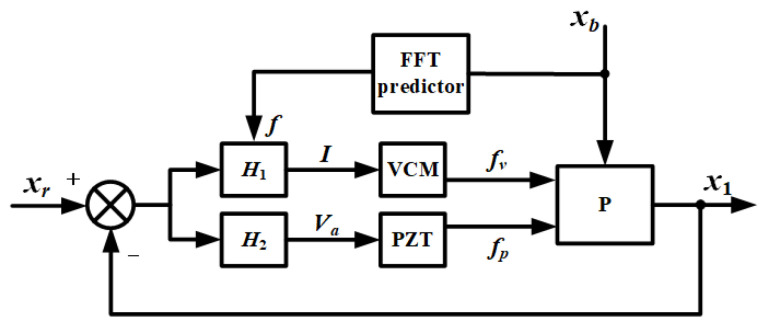
Closed-loop control diagrammatic sketch.

**Figure 8 sensors-21-01941-f008:**
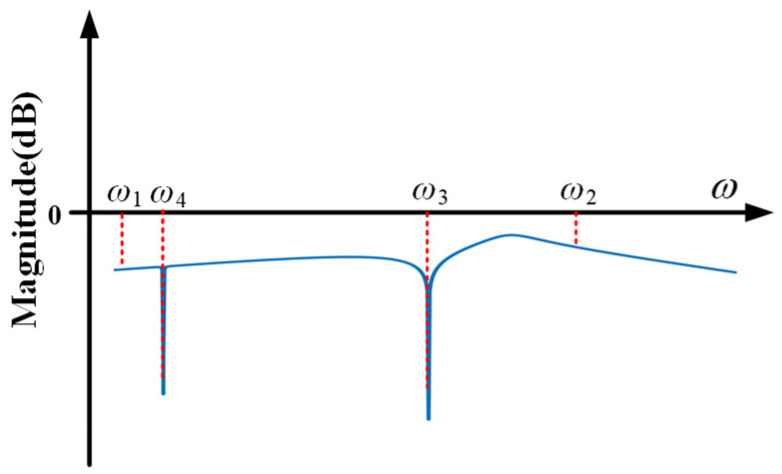
Bode diagram of the system with designed controller.

**Figure 9 sensors-21-01941-f009:**
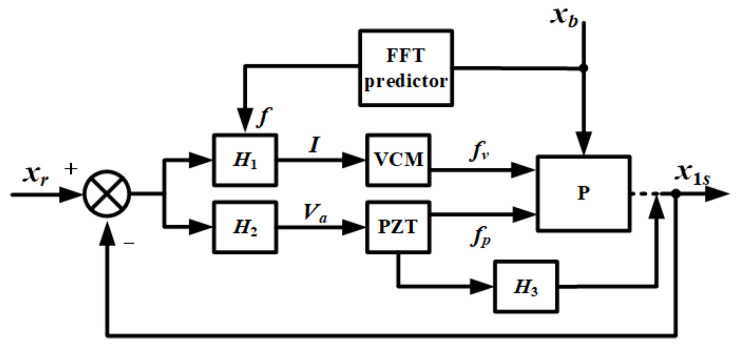
Self-sensing closed-loop control diagrammatic sketch.

**Figure 10 sensors-21-01941-f010:**
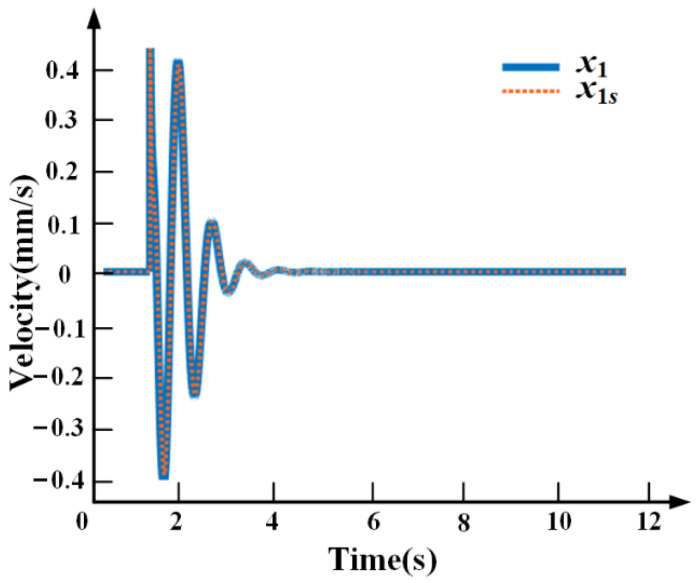
The simulation comparison between the speed sensor and the self-sensing.

**Figure 11 sensors-21-01941-f011:**
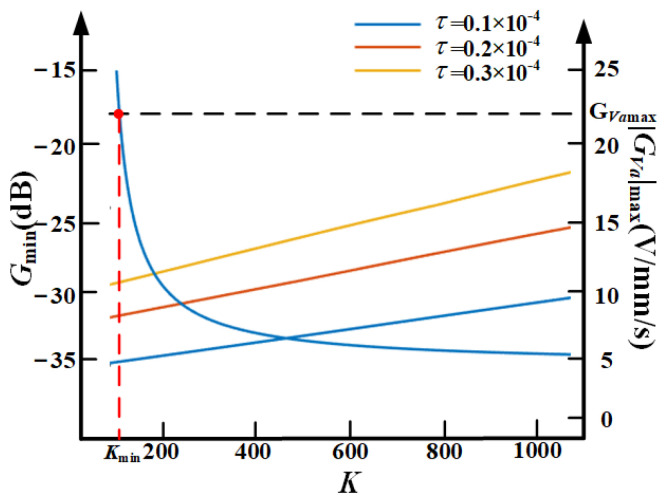
The relationship curves of *G*_max_ and |*G_Va_*|_max_ with *K.*

**Figure 12 sensors-21-01941-f012:**
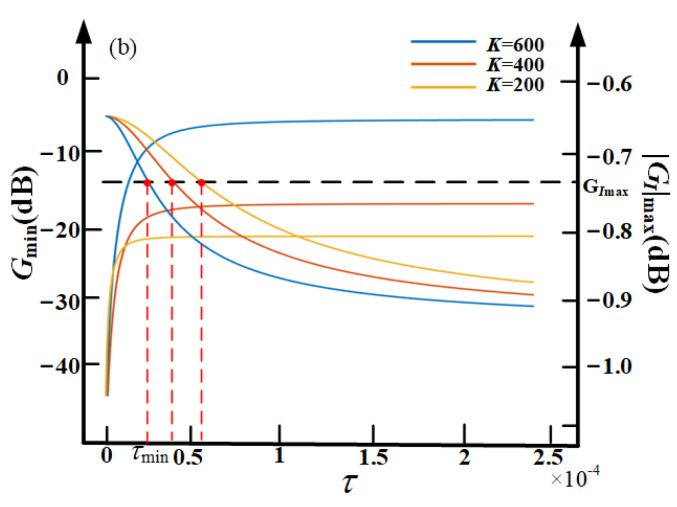
The relationship curves of *G*_max_ and |*G_I_*|_max_ with τ.

**Figure 13 sensors-21-01941-f013:**
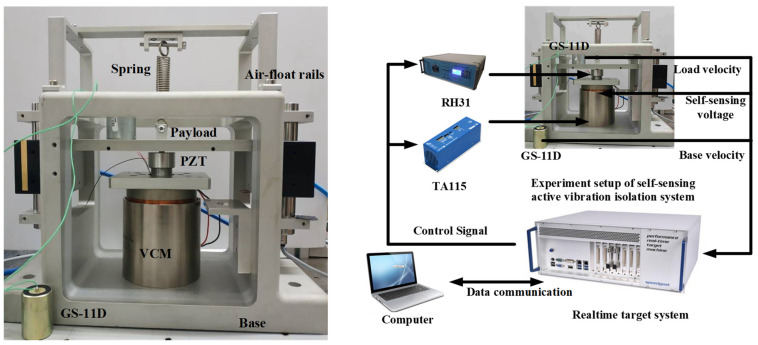
The experiment setup of the self-sensing active vibration isolation system.

**Figure 14 sensors-21-01941-f014:**
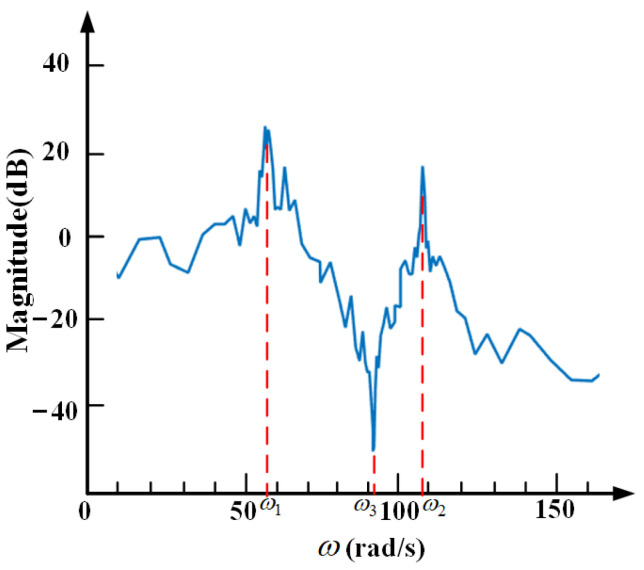
Transmissibility curve of the self-sensing active vibration isolation system without any controller.

**Figure 15 sensors-21-01941-f015:**
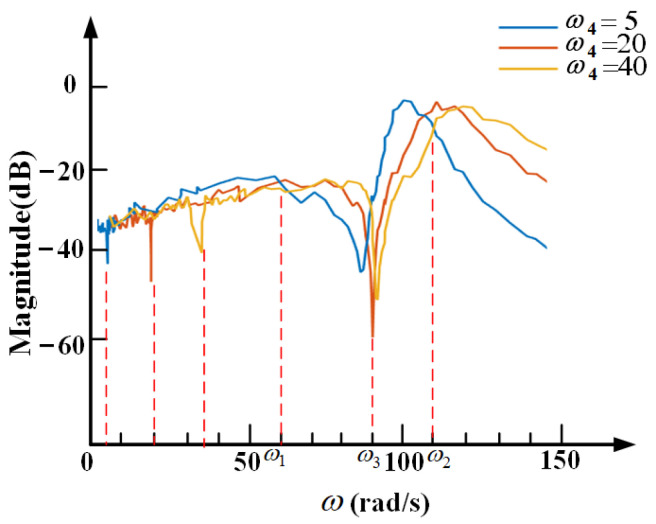
Transmissibility curve with the self-sensing controller.

**Figure 16 sensors-21-01941-f016:**
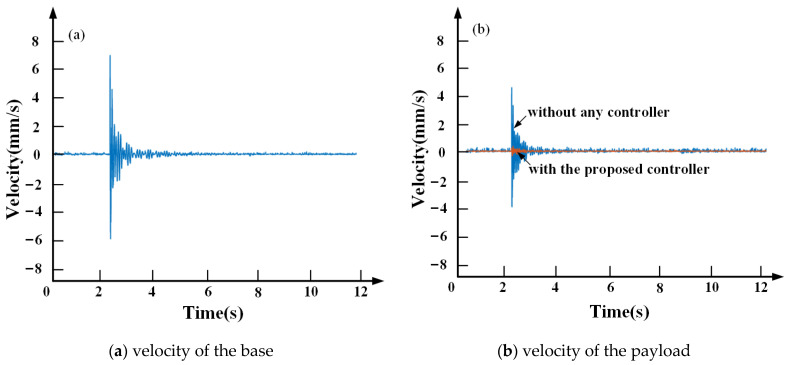
Velocity of the base and the payload.

**Figure 17 sensors-21-01941-f017:**
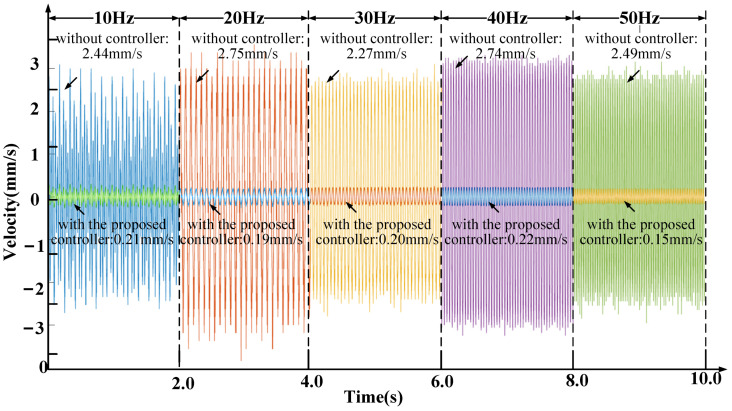
The velocity of payload with frequency sweeping disturbance.

**Figure 18 sensors-21-01941-f018:**
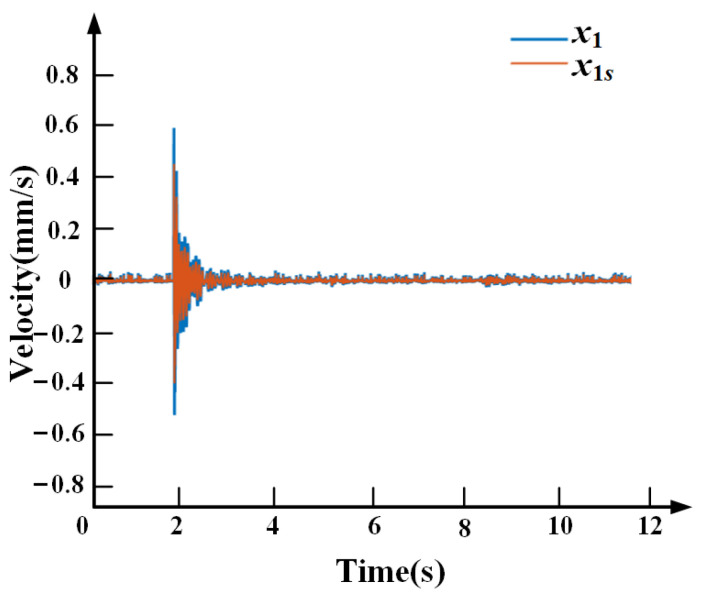
The comparison of vibration performance between the velocity sensor and the self-sensing result.

**Figure 19 sensors-21-01941-f019:**
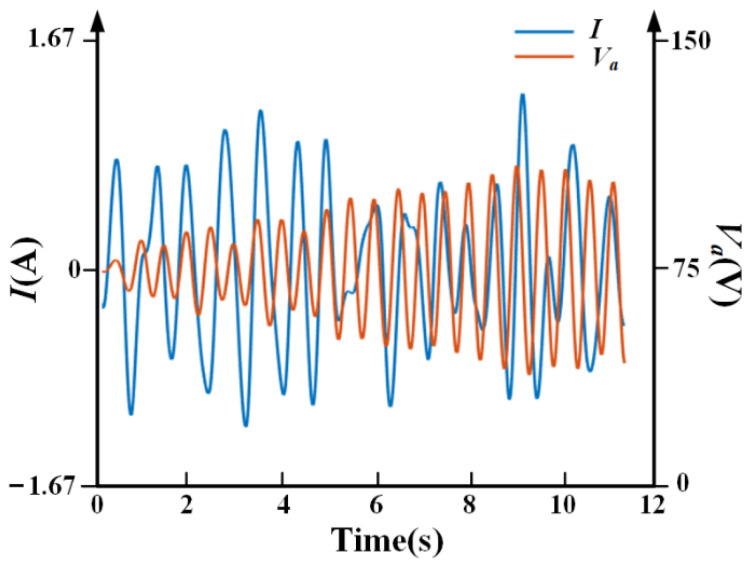
The curves of the control signal *I* and *V_a_.*

**Table 1 sensors-21-01941-t001:** Parameters of the self-sensing active vibration isolation system.

Parameter	Value
Mass of the payload, *m*_1_ (kg)	0.695
Sum mass of the VCM, the PZT, and connecting plate, *m*_2_ (kg)	0.614
Stiffness of the passive vibration isolation system, *k* (N/m)	480
Stiffness of PZT actuator, *k_p_* (N/m)	5000
Thrust coefficient of the VCM, *k_i_* (N/A)	31.8
Thrust coefficient of the PZT, *k_u_* (N/V)	25.4

## Data Availability

Data available on request from the authors.

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
