# Peer review of "Design and Experimentation of a Self-Sensing Actuator for Active Vibration Isolation System with Adjustable Anti-Resonance Frequency Controller"

_sensors, 2021, doi:10.3390/s21061941_

Round 1
Reviewer 1 Report
Please see the PDF file attached

Author Response
Please see the word file attached.

Reviewer 2 Report
Paper is well developed, but why did you not including the leading groups from USA or Europe.
- Check Urbikain, Campa, Budak, Stephan…they proposed new ways of solving equation, by frequency or semi-discretization approaches. Olvera is another guy proposing. In this field 20 references must be from no Asiatic countries.
- IJ sound and vibration is a good source, check about the above cited authors
Self-sensing controller for vibration isolation system with anti-resonance frequency is proposed, that is in use in machine tools after the use of Micromega devices.
Why did you not write the highlights as points in conclusions?
Ref 8 and 9,,,this is about microscopes. Again the same idea, perhaps you are including no direct linked works, check them all.
Fig 17 and 18 can be only one, you change from double to single ones, Fig 15 is interesting
I do not see flwas in the theory…however the state of the art is bad, in simple words
Author Response
Please see the word file attached.

Round 2
Reviewer 1 Report
Please find the attached PDF file.

Reviewer 2 Report
Conclusions: you must rewrite them to make cleared wich ones are the main contributions, point by point.
The way to solve the equations by discretization methods can be checked in the work Stability prediction in Int J Machine Tool and Manuf 54, 73-81 . Semi-descrization methods are key today.
Check final version
